# Half-Life Extension and Biodistribution Modulation of Biotherapeutics via Red Blood Cell Hitch-Hiking with Novel Anti-Band 3 Single-Domain Antibodies

**DOI:** 10.3390/ijms24010475

**Published:** 2022-12-28

**Authors:** Toan D. Nguyen, Brandon M. Bordeau, Yu Zhang, Anna G. Mattle, Joseph P. Balthasar

**Affiliations:** 1Department of Pharmaceutical Sciences, University at Buffalo, Buffalo, NY 14214, USA; 2450 Pharmacy Building, Buffalo, NY 14214, USA

**Keywords:** single-domain antibody, red blood cell hitch-hiking, half-life extension, pharmacokinetics, phage display, bispecific antibody

## Abstract

Small therapeutic proteins are receiving increased interest as therapeutic drugs; however, their clinical success has been limited due to their rapid elimination. Here, we report a half-life extension strategy via strategy via red blood cell red blood cell (RBC) hitch-hiking. This manuscript details the development and characterization of novel anti-RBC single-domain antibodies (sdAbs), their genetic fusion to therapeutic antibody fragments (TAF) as bispecific fusion constructs, and their influence on TAF pharmacokinetics and biodistribution. Several sdAbs specific to the band 3 antigen were generated via phage-display technology. Binding affinity to RBCs was assessed via flow cytometry. Affinity maturation via random mutagenesis was carried out to improve the binding affinity of the sdAbs. Bi-specific constructs were generated by fusing the anti-RBC sdAbs with anti-tissue necrosis factor alpha (TNF-α) TAF via the use of a glycine-serine flexible linker, and assessments for binding were performed via enzyme-linked immunosorbent assay and flow cytometry. Pharmacokinetics of anti-RBC sdAbs and fusion constructs were evaluated following intravenous bolus dosing in mice at a 1 mg/kg dose. Two RBC-binding sdAbs, RB12 and RE8, were developed. These two clones showed high binding affinity to human RBC with an estimated K_D_ of 17.7 nM and 23.6 nM and low binding affinity to mouse RBC with an estimated K_D_ of 335 nM and 528 nM for RB12 and RE8, respectively. Two derivative sdAbs, RMA1, and RMC1, with higher affinities against mouse RBC, were generated via affinity maturation (K_D_ of 66.9 nM and 30.3 nM, respectively). Pharmacokinetic investigations in mice demonstrated prolonged circulation half-life of an anti-RBC-TNF-α bispecific construct (75 h) compared to a non-RBC binding control (1.3 h). In summary, the developed anti-RBC sdAbs and fusion constructs have demonstrated high affinity in vitro, and sufficient half-life extension in vivo.

## 1. Introduction

Biotherapeutics have gained outstanding success in recent years, with more than 200 approved for clinical use and more than 1000 currently in clinical development [1]. In recent years, there has been increased interest in the development and therapeutic use of small (i.e., MW < 50 kDa) polypeptide constructs, including peptides, single domain antibodies (sdAb, a.k.a. nanobodies), single chain variable fragments (scFv), diabodies, Bites, etc. Relative to immune gamma globulin (IgG) monoclonal antibodies (mAb) [2,3], the smaller constructs may offer improved tissue distribution, more rapid absorption (e.g., following subcutaneous administration), more predictable, linear elimination, and, possibly, decreased cost-of-goods. However, the small constructs carry a significant disadvantage relating to their rapid elimination and short in-vivo biological half-life. To mitigate this disadvantage, several strategies have been implemented, including development of Fc-fusion proteins (e.g., peptibodies), conjugation to polyethylene glycol (PEGylation), fusion to repeats of proline-alanine-serine (PASylation), and fusion to domains that bind to IgG or albumin [1,4]. In this work, we pursue a less-investigated strategy for half-life extension: fusion of small constructs to domains with high affinity for red blood cell (RBC) surface proteins.

Band 3 protein (b3p), also known as anion transport protein 1 (SLC4A1), is expressed primarily in the RBC membrane and, to a lesser extent, on the basolateral face of collecting ducts of the nephron [5]. B3p contributes to cell membrane mechanical support through physical linkage to ankyrin and the cytoskeletal network, and regulates pH inside RBC and for urine. It is the most abundant membrane protein in human erythrocytes, with approximately 1 million copies per RBC, making it a desirable target for our approach [6].

The recent discovery of a unique antibody class devoid of light chains in camelids and cartilaginous fish (e.g., sharks) has ignited the interest in developing alternative immunoglobulin scaffolds [7,8]. These heavy-chain-only antibodies have increased surface solubility due to several transmutations in their VH domains (VHH) in the areas typically enfolded by the V_L_ (i.e., in conventional antibodies), leading to good solubility and stability [9]. The VHHs can be isolated and expressed as single-domain antibodies (sdAbs) while still retaining their antigen recognition and have been utilized extensively in analytical, diagnostic, and therapeutic applications [10,11,12]. This manuscript details the development, characterization, and pharmacokinetic studies of anti-RBC sdAbs and fusion constructs.

## 2. Results

### 2.1. RBC Immunization, Panning, and Screening

In preparation for llama immunization, bio-panning, and screening, RBC b3p was extracted from human RBC. SDS-PAGE gel analysis confirmed a prominent band of the b3p at around 95 kDa with an acceptable purity (Figure 1A). Each 10 mL of 10% human RBC commercial solution yields approximately 500 µg of extracted b3p. A sdAb phage display library was built from isolated PBMCs of a llama immunized with the extracted b3p, panned against human RBC, and screened with ELISA and flow cytometry. As shown in Figure 1B, eight positive sdAb clones binding to the b3p extracted from human RBC were identified via ELISA, and among them, only RE8 and RB12 clones showed binding activity to mouse RBC.

### 2.2. Characterizations of Binding Affinity and In Vivo Pharmacokinetics of RB12 and RE8

The binding affinity RB12 and RE8, the two anti-RBC sdAb clones that showed binding to both human and mouse RBC, was determined via flow cytometry. Both clones showed high binding affinity to human RBC with estimated K_D_ values of 17.7 nM and 23.6 nM for RB12 and RE8, respectively. The binding affinity to mouse RBC was lower than to human RBC, with estimated K_D_ values of 335 nM and 528 nM (Figure 2A).

The observed concentration-time profiles of RB12 and RE8, after an IV administration of 1 mg/kg with a tracer dose of ^125^I labeled sdAb, are shown in Figure 2B. RBC and plasma were separated from whole blood, TCA precipitated, and radioactivity was counted. Plasma half-life improved from 30 min for the control (i.e., non-RBC binding) sdAb 1HE to 5 h for the two RBC-binding sdAbs RE8 and RB12.

### 2.3. Affinity Maturation of RB12 and RE8

A derivative library was constructed from RB12 and RE8 via random mutagenesis, and affinity maturation was performed through additional panning and screening procedures to isolate clones with higher affinity to mouse RBC. Several clones were identified, the majority sharing a common single-point mutation at E109 to lysine. Further binding assessments of two clones, RMA1, and RMC1, were carried out via flow cytometry. No binding to human RBC was detected up to concentrations of 1 µM of the purified sdAbs, and binding affinity to mouse RBC was substantially improved, with estimated K_D_ values of 66.9 nM and 30.3 nM for RMA1 and RMC1 (Figure 3A).

Due to the higher affinity, RMC1 was selected as the lead candidate for subsequent studies. The observed concentration-time profiles of RMC1, after an IV administration of 1 mg/kg with a tracer dose of ^125^I labeled sdAb, are shown in Figure 3B. This dose level was selected as it is within the linear range of binding (i.e., no saturation of binding). RBC and plasma were separated from whole blood, TCA was precipitated, and radioactivity was counted. Plasma half-life was significantly improved from 5 h of the parent sdAbs RB12 and RE8 to 60 h of the affinity maturated sdAb RMC1.

### 2.4. Development and Assessment of Anti-RBC-aTNF Fusion Bispecific Antibody

An RBC-binding fusion construct was developed and characterized. RMC1, or a non-binding control sdAb 1HE, was linked to an anti-tumor necrosis factor alpha (TNF-α) TAF via a flexible glycine-serine (G_4_S)_3_ linker. The bispecific constructs have molecular weights of approximately 33 kDa (Figure 4A), and binding affinity K_D_ to mouse RBC was estimated to be 63.9 nM via flow cytometry (Figure 4B). In addition, the bispecific construct also exhibited an estimated binding IC_50_ of 307 pM to recombinant human TNF-α via ELISA (Figure 4C).

### 2.5. Pharmacokinetics and Biodistribution of the Anti-RBC-TNF-α Fusion Antibody

The observed systemic concentration-time profiles of RMC1-aTNF and 1HE-aTNF, after an IV administration of 1 mg/kg with a tracer dose of ^125^I labeled construct, are shown in Figure 5A. RBC and plasma were separated from whole blood, TCA was precipitated, and radioactivity was counted. Plasma half-life improved from 1.3 h for the control (i.e., non-RBC binding) construct 1HE-aTNF to 75 h for the RBC-binding RMC1-aTNF construct. Improvement in systemic PK also leads to increased tissue exposures of RMC1-aTNF compared to 1HE-aTNF, especially in muscle, liver, spleen, heart, lungs, and kidneys (Figure 5B).

### 2.6. Hematological Analysis of RMC1-aTNF in Mice

To further analyze the safety of treatment with a high dose (5 mg/kg) of RBC-binding sdAbs, hematological analysis was performed in mice. As shown in Figure 6, minimal impact of treatments with low-affinity or high-affinity anti-RBC sdAbs on hematological parameters and body weight was observed compared to the control group, indicating that the construct does not produce significant hematologic toxicity at the administered dose.

## 3. Discussion

RBCs have been utilized previously in drug delivery strategies due to their unique and convenient properties, including homogeneity, abundance, long circulation half-life, remarkable endurance and resilience to external modification, and lack of nuclei and intracellular organelles [14]. There are two main drug-loading approaches applied in RBC delivery systems. The first approach employed the encapsulation of therapeutic agents into the intracellular space of the RBC via an osmotic loading procedure [15]. The encapsulation procedure involves pore opening via exposure of RBCs to hypotonic conditions, and the therapeutic agents present in the solution are allowed to diffuse passively into RBCs for loading. Physiological osmolarity is then restored, which induces pore closing and entrapment of the loaded agents inside the RBCs. The second approach pursues attachment of the therapeutic cargos to the RBC membrane via chemical conjugations, direct absorption, genetic modification, biotinylation, and/or with fusion conjugates with an affinity for RBC surface antigens [14,15]. Of note, among the extracellular RBC loading methods, the use of affinity ligands targeting RBCs is the only method that does not require ex vivo manipulation and reinfusion of RBCs [14].

Initial efforts aimed to target surface antigens for coupling therapeutics to RBCs targeted complement receptor type 1 (CR1), which functions as an anchorage site for immune complexes containing activated complement C3b [16]. Several applications have shown increased benefits of therapeutic agents when coupled to anti-CR1 antibodies, in vitro, and in animal models. For example, conjugating tissue plasminogen activator, an enzyme involved in the breakdown of blood clots, to an anti-CR1 mAb extended its circulating half-life and improved its prophylactic thrombolytic application [17]. Similarly, coupling mAbs against pathogens and botulinum neurotoxin to anti-CR1 moieties has shown greater neutralizing potency than un-modified mAbs [18,19,20,21,22]. A significant drawback with CR1 is its low expression level in RBCs, with only 500–1500 copies per cell. The expression level of CR1 is significantly below the capacity needed, and thus was considered to be unsuitable for a half-life extension strategy.

Glycophorin A (GPA) has also been explored as an RBC surface protein target for use in half-life extension strategies. GPA is a membrane-bound sialoglycoprotein of the human erythrocyte membrane, which contributes to RBC structural integrity and surface charge [23]. The expression level of GPA in RBCs is significantly higher than CR1, with around 800,000 copies/cell, making it a more attractive target. Anti-GPA antibodies have been employed to make fusion constructs with human thrombomodulin and tissue plasminogen activator for antithrombotic therapies [24,25,26]. However, antibodies binding to GPA increase RBC membrane rigidity [27], altering RBC physiology and functionality, leading to stress-induced hemolysis and the generation of reactive oxygen species [27,28].

B3p is the most abundant RBC membrane protein with 1.2 million copies/cell, accounting for 25% of the total membrane protein. It is expressed selectively in RBC [6,29]; in addition, b3p is highly conserved across different populations [30]. Unfortunately, the binding of b3p with monoclonal antibodies showed similar negative effects on RBC membrane integrity as observed with antibodies binding to GPA. However, monovalent binding via Fab was shown to lead to little impact on membrane elasticity [31] and, consequently, b3p is a very promising target for our RBC-binding strategy.

In the Camelidae family, IgG antibodies consist of three isotypes: IgG1, which has the conventional structure composed of both light chains and heavy chains, and the heavy-chain only IgG2 and IgG3 [11]. Due to having only a single binding domain, cloning llama VHH repertoires to a phage display library fully retains the diversity of heavy-chain IgG repertoires [11]. Furthermore, sdAb isolated from the VHH libraries has several advantages over scFv and Fab, including better stability, higher expression yield, less propensity to form dimers and aggregates, and being smaller and easier to engineer [10,12,32,33]. This study prepared a sdAb phage display library from an immunized llama to develop anti-b3p sdAbs. Compared to the heavy chain of IgG1, both IgG2 and IgG3 are shorter due to the lack of a CH1 domain; therefore, amplified DNA of VHs and VHHs can be separated by size via gel electrophoresis. The amplified VHH genes were cloned in the appropriate phagemid vector to construct the sdAb phage library.

In several rounds of panning and screening, several sdAb clones were identified that bind to human b3p and human RBC, but only 2 of those clones, RE8 and RB12, also bound to mouse RBC. Since we wished to demonstrate proof-of-concept studies in mice, these two clones were selected for further characterizations and optimizations. Assessment via flow cytometry showed that both sdAbs had high affinity against human RBC but low affinity against mouse RBC. Clones with higher affinity to mouse RBC were obtained through mutagenesis and the affinity maturation process; however, enhanced binding to mouse RBC was accompanied by loss of binding to human RBC. Although it would be desirable for constructs to have high binding to both mouse and human RBC, the species selectivity of RBC binding is consistent with binding selectivity of the sdAbs to b3p. The lead sdAb clone RMC1 was selected to generate anti-RBC fusion constructs.

Small therapeutic proteins have been developed for many therapeutic applications, including T-cell retargeting [34], neutralization of soluble toxins [35,36], and masking antibody and antibody-conjugate binding sites to mitigate the binding-site barrier [37,38,39]. Although small proteins have demonstrated utility in animal models and in humans, their clinical use may be limited, at least to some extent, by their rapid systemic clearance and short biological persistence. Several half-life extension strategies have been developed, including fusion with albumin-binding domain and Fc receptor or chemical modifications such as PEGylation and lipidation [40,41,42,43,44]. However, RBC as a carrier for half-life extension strategy has several advantages, including restricted presence in the vascular space, a long life span of 120 days, and a slow turnover rate of less than 1% per day [45]. Our PK study in mice revealed that anti-RBC sdAbs were strongly associated with RBC, and circulation half-lives were prolonged from 30-min for the non-binding sdAb 1HE up to 60 h for anti-RBC sdAb RMC1. RBC-binding also led to a half-life extension of up to 75 h with the anti-RBC-TNF-α construct compared to 1.3 h with the non-binding construct.

In conclusion, the present work developed and evaluated anti-RBC sdAb and bispecific constructs. The anti-RBC sdAb exhibited slow clearance from the systemic circulation, which is a desirable attribute for our half-life extension strategy. Future investigations will evaluate the benefits of the anti-RBC strategy in improving the therapeutic activity of small protein constructs.

## 4. Materials and Methods

### 4.1. Extraction of Red Blood Cell B3p

B3p was extracted from human RBC via a previously described method [46]. Human RBC (Rockland Immunochemicals, Pottstown, PA, USA) were washed two times with ten volumes of PBS pH 7.4 and centrifuged at 3000× *g* for 5 min. The washed cells were mixed with ten volumes of ice-cold 5P8 buffer (5 mM sodium phosphate, pH 8.0) containing 0.2 mM DTT, 20 ug/mL PMSF, and incubated for 10 min on ice. The cell lysate was centrifuged for 20 min at 27,000× *g* at 4 °C, and hemolysate was removed by aspiration. The ghost cell pellet was washed for an additional four times with 5P8 buffer.

The ghost cell membrane was washed with 10 vol of ice-cold SE buffer, then centrifuged at 46,000× *g* for 20 min at 4 °C. The pellet was resuspended with a syringe in 10 vol of SE buffer, incubated at 37 °C for 30 min, then centrifuged at 46,000× *g* for 20 min at 4 °C. This process was repeated one more time before resuspending the spectrin-depleted ghost membrane with a syringe in 10 vol of KI extraction buffer and incubating for 30 min at 37 °C. The ghost membrane was pelleted by centrifugation for 30 min at 46,000× *g* at 4 °C, then resuspended in 5P8 + PMSF buffer, and incubated at 4 °C for 10 min. The ghost membrane was centrifuged at 46,000× *g* for 30 min and washed once more time with 5P8 + PMSF buffer. To solubilize the KI-extracted ghosts, 4 vol of 1% C_12_E_8_ (*v*/*v*) in 228 mM sodium citrate pH 8.0 + 1 mM DTT + PMSF was added, and incubated for 20 min on ice. The resulting suspension was centrifuged at 46,000× *g* for 30 min (or up to 80,000× *g* for 45 min) at 4 °C, and the supernatant was collected and further purified with affinity chromatography following published procedures [47]. B3p isolation and purity were confirmed with sodium dodecyl sulphate-polyacrylamide gel electrophoresis (SDS-PAGE), which demonstrated protein isolation with the expected molecular weight of 95 kDa.

### 4.2. Red Blood Cell Antigen Immunization and Phage-Display Library Construction

To establish a single domain antibody (sdAb) phage display library, a llama provided by Capralogics (Hardwick, MA, USA) was subcutaneously immunized with 300 μg of extracted human b3p in incomplete Freund’s adjuvant every three weeks for a total of four immunizations. Ten days after the fourth immunization, 600 mL of freshly harvested blood was used to build a phage-displayed library as described previously [48].

### 4.3. Phage Panning to Enrich sdAb Binders

To enrich positive binders, 50, 25, and 10 µL of 10% human RBCs were diluted in PBS buffer for the 1st, 2nd, and 3rd rounds of panning, respectively. Cells were washed three times with PBS and blocked with MPBS (PBS + 5% non-fat dry milk) at RT for one h. Following washing with PBS, for the first panning input, the stock phage was diluted to 10^12^ c.f.u. (colony forming unit)/mL in blocking buffer (2% milk PBS), and 100 µL of the diluted phage was distributed into each well and incubated for 2 h at room temperature. For the subsequent panning, the stock phage was diluted 1:1 in the blocking buffer. After the incubation, the plate was washed with PBST (PBS + 0.05% Tween 20) 5 times, 10 times, 15 times, and 15 times for the 1st, 2nd, 3rd, and 4th rounds of panning. Bound phages were then eluted by incubation with 100 µL of 1 mg/mL trypsin for 30 min at room temperature. Output phage was titrated and re-infected with TG-1 cells for phage production in the subsequent round of panning.

### 4.4. Screening of sdAb Binders

Phage-infected TG1 cells were grown overnight, serially diluted in LB, spread over individual culture plates containing selective medium (LB agar + 100 µg/mL ampicillin + 2% (*w*/*v*) glucose), and incubated overnight at 37 °C. A master plate was generated by inoculating a single colony into wells of a 96-well round-bottom culture plate filled with 100 µL of 2xYT supplemented with 100 µg/mL ampicillin, 2% (*w*/*v*) glucose, and 10% (*v*/*v*) glycerol, and grown overnight at 37 °C. These wells were then used to inoculate wells of 96-deep well plates containing 1 mL of 2xYT medium (100 µg/mL ampicillin + 0.1% (*w*/*v*) glucose per well). Plates were incubated for three hours at 37 °C and 250 rpm until OD600 ≈ 0.5. Nanobody expression was induced with 4 µL of 0.5 M IPTG per well (final concentration of 2 mM) and incubated for an additional 15–24 h at 37 °C and 300 rpm.

Nunc Maxisorp 96-well plates were coated with 0.4 µg/well of extracted human b3p overnight at 4 °C. Plates were washed three times with PBST and blocked with MPBS for two hours at RT. Plates were then washed three times with PBST and incubated with a 1:1 diluted supernatant or cell lysates from induced bacteria at RT for two hours. Plates were then washed three times with PBST, where bound sdAb was detected using an anti-HA tag HRP-conjugated Ab diluted 1:1000 in PBST. Following incubation and five washings with PBST, 100 μL of 1-Step Turbo TMB-ELISA solution was added to each well and incubated for 30 min. The reaction was quenched by adding 100 µL of Stop Solution to each well, and absorbance was measured at 450 nm.

### 4.5. Protein Expression and Purification

An anti-TNF-α sdAb sequence taken from Beirnaert et al. [13] was fused with the the anti-RBC sdAb RMC1 or with a non-binding control (1HE) sdAb with use of a flexible glycine-serine (G_4_S)_3_ linker. The DNA of sdAbs and bispecific constructs was ligated to a cut expression plasmid PET-22b and transformed to the *E. Coli* Shuffle T7 cells. A single colony was inoculated, and overnight grown culture was diluted 1:100 to TB medium. The cells were grown at 30 °C until OD600 = 0.6–0.8, and 1 mM IPTG (Enzo Life Sciences, Farmingdale, NY, USA) was added to the culture to induce expression. After a 20-h expression, the protein in cell pellets was extracted by Bugbuster buffer (MilliporeSigma, Burlington, MA, USA) and purified by Ni-NTA resin (Thermo Fisher Scientific, Rockford, IL, USA) by following the manufacturer’s protocol. The sdAb was then dialyzed into loading buffer (10 mM Na_2_HPO_4_, 5 mg/L CaCl_2_, pH 6.5) and purified using a Bio-Scale Mini Ceramic Hydroxyapatite Multimodal Chromatography Type I (CHT) Cartridge (Bio-Rad, Hercules, CA, USA). Briefly, the CHT resin was pre-equilibrated with 3 column volumes (CV) of elution buffer (500 mM Na_2_HPO_4_, 5 mg/L CaCl_2_, pH 6.5) and then equilibrated with 10 CV of loading buffer at a rate of 2.0 mL/min. The sample was then loaded onto the column at a rate of 2.0 mL/min followed by 5 CV of loading buffer. The elution process consisted of gradients at a rate of 2.0 mL/min of 20 CV of 0–100% elution buffer. Fractions were collected during the elution process based on an A280 ≥ 0.05. Antibody purity was assessed by SDS-PAGE analysis. Briefly, antibody samples were prepared in non-reducing buffer with boiling for 5 min. Samples were then loaded on an SDS-PAGE gel (Bio-rad), and electrophoresed at 120 V for 30 min. The gel was stained with Coomasie blue R-250 (Bio-Rad) and destained with double distilled water following the manufacturer protocol, and the migration distances of different proteins were compared to the migration distances of molecular weight standards.

### 4.6. Flow Cytometry Analysis

ELISA-positive clones (RA1, RA8, RB12, RD1, RD11, RE8, RG10, and RH5) were tested for binding to red blood cells via flow cytometry. Briefly, whole blood from Swiss Webster mice (Taconic Biosciences, Rensselaer, NY, USA) was collected in ethylenediaminetetraacetic acid (EDTA) and spun at 1000× *g* for 10 min, and plasma and the buffy layer were discarded. The remaining mouse RBCs were washed thrice and resuspended in PBS (500× *g* 10 min). The commercial human or isolated mouse RBCs were counted and washed once with PBS + 1% Bovine serum albumin (BSA), and 1 × 10^6^ cell aliquots were dispensed into each 5 mL flow cytometry tube (VWR, Bridgeport, NJ, USA). Cells were then pelleted by centrifugation (300× *g* for 5 min), resuspended into 300 µL of induced cell culture medium, and incubated for 1 h at 4 °C. Cells were washed two times with 1 mL of washing buffer each time and then incubated with 300 µL of washing buffer containing mouse anti-His-tag secondary antibody (Thermo Fisher Scientific, Rockford, IL, USA) diluted 1:500 *v*/*v* for one hour at 4 °C. To label the antibodies, 250 µL of PE-conjugated goat anti-mouse polyclonal antibodies were diluted at 1:250 *v*/*v* for 30 min at 4 °C. Samples were washed, resuspended in 500 µL of washing PBS on ice, and protected from light until flow cytometry analysis (BD Fortessa SORP, BD Biosciences, San Jose, CA, USA). Purified anti-RBC sdAbs or fusion constructs at a range of concentrations from 0.03 nM to 1000 nM were analyzed to generate binding curves, and binding affinity (*K_D_*) was estimated using GraphPad Prism software (San Diego, CA, USA) using the equation below:$$Y=B_{max}\times X^{h}/\left(K_{D}^{h}+X^{h}\right)$$
where *B_max_* is the binding capacity, *K_D_* is the binding affinity, and *h* is the Hill slope.

### 4.7. Radiolabeling of Antibodies

Antibodies were radiolabeled with iodine-125 (^125^I) following a modified chloramine-T method described previously [49]. In short, 40 µL of purified antibodies at 1–2 mg/mL concentration in pH 7.4 PBS was reacted with 10 µL of 100 mCi/mL sodium ^125^I (PerkinElmer, Waltham, MA, USA) and 20 µL of 1 mg/mL chloramine-T for 90 s, then 40 µL of 10 mg/mL potassium iodide was added to terminate the reaction. Subsequently, ^125^I labeled antibodies were separated from the mixture via gel filtration (Sephadex G-25 column, GE Healthcare Bio-Sciences, Pittsburgh, PA, USA). Purity and radioactive activity of the ^125^I-antibodies were assessed via thin layer chromatography (PE SiL-G, Whatman Ltd., Kent, UK) and gamma counting (LKB Wallac 1272, Wallac, Turku, Findland), respectively.

### 4.8. Pharmacokinetics of Anti-B3p SdAbs

For the pharmacokinetic studies of the anti-band-3 sdAbs (RE8, RB12, and RMC1), doses of 1 mg/kg of unlabeled antibodies and 400 µCi/kg tracer doses of iodinated antibodies were administered via a penile vein to three homozygous Nu/J mice (The Jackson Laboratory, Bar Harbor, ME, USA) or Swiss Webster mice. Blood was collected from the retro-orbital sinus, and blood samples (50 µL/sample) were centrifuged at 200× *g* for five minutes to separate plasma and RBCs. The samples of RE8 and RB12 were collected at 5 min, 1, 3, 6, 9, and 12 h post-dosing. The samples of RMC1 were collected at 1, 3, 6, 12, 24, and 48 h post-dosing. Plasma (20 µL/sample) and RBCs (~25 µL/sample) were separately precipitated with trichloroacetic acid (TCA). BSA (200 µL of 1%) in PBS and TCA (700 µL of 10%) were added to samples, followed by 15-min incubation on ice. The precipitate was isolated by centrifuging at 14,000 rpm for five minutes and discarding the supernatant. The precipitate was then washed three times with PBS. Radioactivity from plasma and RBC TCA precipitation was counted using a gamma counter and corrected for background radioactivity, gamma counter efficiency, and radioactive decay. Free and RBC-bound antibody concentrations were determined from plasma radioactivity and RBC radioactivity, respectively. Whole blood antibody concentration was calculated, assuming a hematocrit of 44.6%. Non-compartmental analysis was performed in Simbiology (Matlab software).

### 4.9. Hematology Study

For the hematology study of anti-RBC sdAbs RMC1 and RB12, Swiss Webster mice were dosed at 5 mg/kg via retro-orbital injection (*n* = 5). Blood was sampled through retro-orbital collection prior to dosing for the determination of baseline measurements of hematological parameters. Additional samples were obtained at several time-points up to 28 days post-dosing. Blood samples were collected using EDTA-coated capillary pipette tubes. Red blood cell concentration, hemoglobin concentration, and hematocrit were determined using a VetScan HM5 Hematology Analyzer (Abaxis Inc., Union City, CA, USA), normalized by the baseline measurements, and reported as a percentage of pretreatment values. In addition to hematological parameters, body weight change was also monitored and analyzed.

### 4.10. Plasma and Tissue Biodistribution of Bispecific Fusion Constructs

For the biodistribution study of RMC1-aTNF and 1HE-aTNF (served as a control), Swiss Webster mice were dosed at 1 mg/kg through IV administration with a tracer dose of ^125^I-Ab at 400 uCi/kg. Blood was sampled through cardiac puncture. Tissue samples were harvested from the heart, lungs, liver, spleen, kidneys, gastrointestinal tract, skin, and muscles. The samples of RMC1-aTNF were collected at 1, 8, 24, 72, 120, and 168 h post-dosing. The samples of 1HE-aTNF were collected at 0.25, 0.5, 1, 1.5, 2, and 4 h post-dosing. For the preparation of the blood samples, whole blood was centrifuged at 300× *g* for 5 min to separate RBC from plasma. For the preparation of the tissue samples, tissues were incubated with 1% BSA in PBS of 10 times the volume for 1 h. RBC samples, plasma samples, and tissue sample wash buffer (1% BSA in PBS) were then TCA-precipitated and radioactivity counted with corrections for background radioactivity, gamma counter efficiency, and radioactive decay. It is assumed that the ratio of free vs. protein-labeled radioactivity in the wash buffer is the same as that in the tissue.

## Figures and Tables

**Figure 1 ijms-24-00475-f001:**
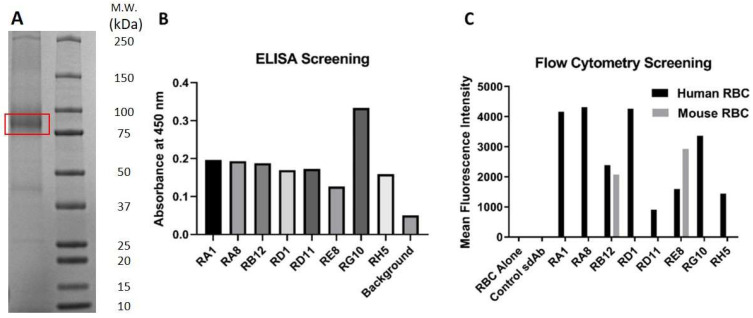
Development of Anti-RBC Single-Domain Antibodies: (**A**) SDS-PAGE analysis of extracted human band 3 protein (b3p) from RBC used for immunization and screening. Extracted b3p from human RBC was exhibited as a prominent band at around 95 kDa with an acceptable purity. The other faint bands are possibly the proteolytic fragments or multimers of b3p. A llama sdAb phage display library was built from PBMCs of a llama immunized with the extracted b3p, panned against human RBC, and screened with ELISA and flow cytometry. (**B**) Assessment of the anti-RBC antibodies binding to the extracted human RBC b3p via ELISA. Eight clones were identified that had a binding signal greater than 3-fold of the background signal. (**C**) Assessment of the anti-RBC antibodies binding to human and mouse RBC by flow cytometry. Human and mouse RBCs were incubated with cell culture media from positive clones. PE-labeled anti-His-tag secondary antibody was used for detection. The calculated mean fluorescence intensity was plotted to quantify binding activity. All of the eight sdAb clones showed binding activity to human RBC, while only RE8 and RB12 showed binding activity to mouse RBC.

**Figure 2 ijms-24-00475-f002:**
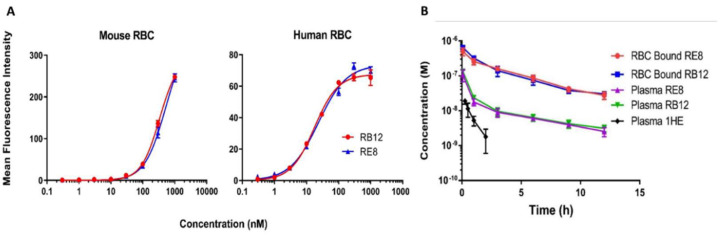
Characterization of Binding Affinity and Pharmacokinetics of Anti-RBC Single-Domain Antibodies. (**A**) The binding affinity of the anti-RBC RB12 and RE8 clones was assessed via flow cytometric analysis. Human and mouse RBCs were incubated with purified sdAbs. Binding activity was detected with a phycoerythrin-conjugated anti-His tag and quantified as mean fluorescence intensity (MFI). Data analysis was carried out using FlowJo software, and the calculated MFI was fitted with a specific binding function with Hill slope in GraphPad Prism software. Both clones showed high binding affinity to human RBC with the estimated K_D_ of 17.7 nM and 23.6 nM for RB12 and RE8, respectively. The binding affinity of these clones to mouse RBC was revealed to be lower than to human RBC, with the estimated K_D_ of 335 nM and 528 nM for RB12 and RE8, respectively. (**B**) The concentration-time profile after an IV administration of 1 mg/kg of RB12 and RE8 with a tracer dose of ^125^I labeled is shown. RBC and plasma were separated from whole blood, TCA was precipitated, and radioactivity counted. Plasma PK data of a control sdAb with no known binding to mouse proteins (1HE) from a previous study in our lab was overlaid. Blood concentrations of the constructs were determined through gamma counting. Relative to 1HE, both RE8, and RB12 exhibited substantially improved persistence in plasma. Significant fractions of circulating RE8 and RB12 sdAbs are bound to RBC. Data points represent the mean concentration (*n* = 3), and error bars denote the standard deviation of the mean.

**Figure 3 ijms-24-00475-f003:**
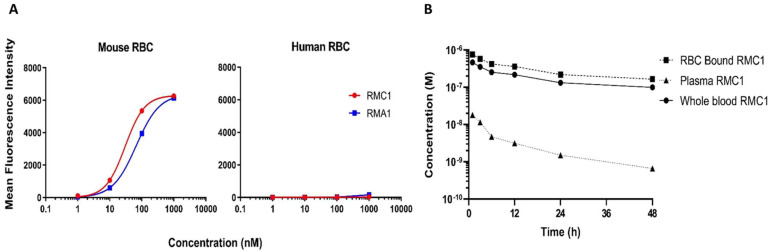
Pharmacokinetics of anti-RBC sdAbs (RMC1) in mice: (**A**) A derivative library was constructed from RB12 and RE8 via random mutagenesis, and affinity maturation was performed through additional panning and screening procedures to isolate clones with higher affinity to mouse RBC. Further binding assessment of two derivative clones, RMA1 and RMC1, was carried out via flow cytometry. There is no binding detected to human RBC up to 1 µM of purified sdAbs, while improved binding affinity to mouse RBC was observed with the estimated K_D_ of 66.9 nM and 30.3 nM for RMA1 and RMC1, respectively. (**B**) The concentration-time profile after an IV administration of 1 mg/kg of RMC1 with a tracer dose of ^125^I label is shown. RBC and plasma were separated from whole blood, TCA precipitated, and radioactivity counted. Plasma half-life improved from 5 h for the two low-affinity RBC-binding sdAbs RE8 and RB12 to 60 h for the affinity-maturated sdAb RMC1. Data points represent the mean concentration (*n* = 3), and error bars denote the standard deviation of the mean.

**Figure 4 ijms-24-00475-f004:**
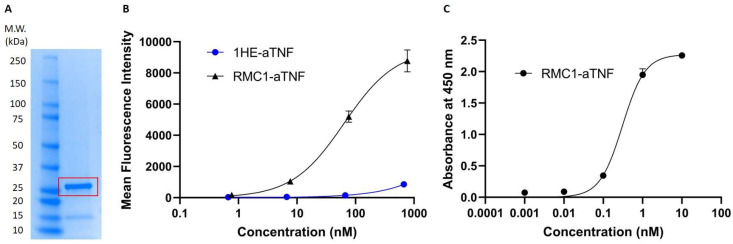
Assessment of binding activity of anti-RBC fusion constructs: (**A**) SDS-PAGE analysis of the anti-RBC-TNF-α RMC1-aTNF bispecific antibody. (**B**) Binding assessment via flow cytometric analysis of purified bispecific sdAb fusion RMC1-aTNF to mouse red blood cells. The anti-RBC-TNFα bispecific sdAb fusion protein was able to retain the RBC binding affinity of RMC1, with an estimated K_D_ of 63.9 nM. The control bispecific sdAbs 1HE-aTNF has some degree of nonspecific binding at the very high concentration of 1 μM. (**C**) Binding assessment via direct ELISA of purified bispecific sdAb fusion RMC1-aTNF to human TNFα. The sdAb aTNF was originally reported to have a high binding affinity to human TNFα with KD around 130 pM [13]. The anti-RBC-TNFα bispecific sdAb fusion protein was able to retain most of this binding affinity, with an estimated KD of 307 pM.

**Figure 5 ijms-24-00475-f005:**
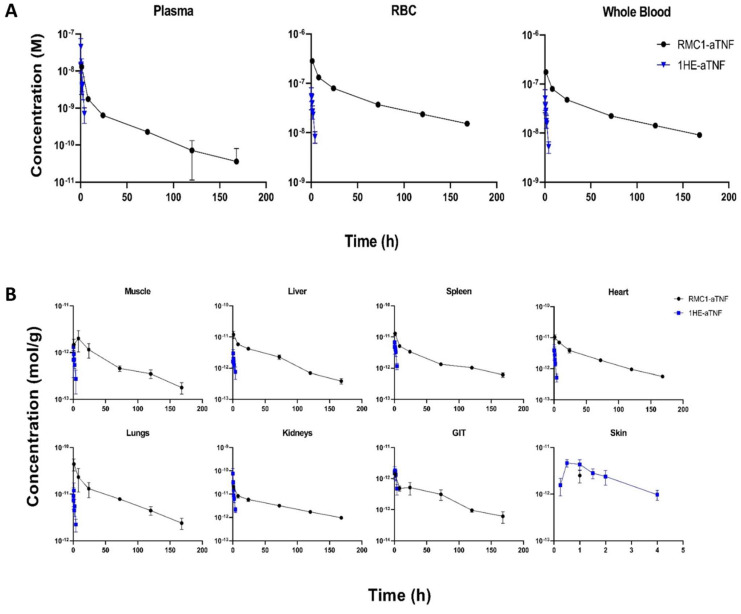
Pharmacokinetics and biodistributions of anti-RBC-TNFα bispecific sdAb fusion protein (RMC1-aTNF) in mice. (**A**) The systemic concentration-time profiles after an IV administration of 1 mg/kg of RMC1-aTNF and 1HE-aTNF with a tracer dose of ^125^I labeled in plasma, RBC, and whole blood are shown. The plasma half-life of RMC1-aTNF was around 75 h, significantly longer than the control bispecific fusion protein 1HE-aTNF (1.3 h). (**B**) The concentration-time profiles of RMC1-aTNF and 1HE-aTNF control in tissues are shown. Data points represent the mean concentration (*n* = 3), and error bars denote the standard deviation of the mean.

**Figure 6 ijms-24-00475-f006:**
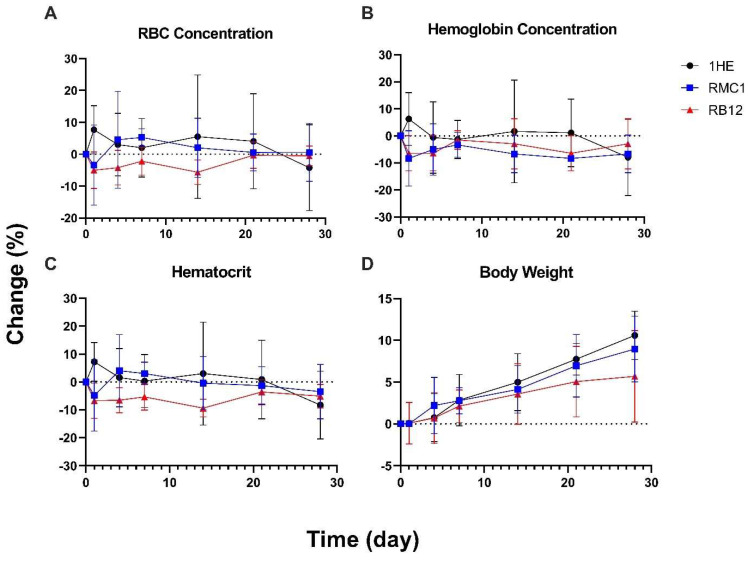
Hematology study of anti-RBC sdAbs. Two-way repeated measures ANOVA was utilized to evaluate the effect of time, treatment (administration of anti-RBC sdAbs), and their interaction on (**A**) RBC concentration, (**B**) hemoglobin concentration, (**C**) hematocrit and (**D**) mouse body weight. There seems to be significant interaction between the effect of time and treatment with RB12 on hematocrit (*p* = 0.0155). Otherwise, the administration of RMC1 and RB12 at the high dose of 5 mg/kg did not result in a significant change in the hematological parameters and body weight in Swiss Webster mice compared with the control sdAb 1HE. Data points represent the mean plasma concentration (*n* = 3), and error bars denote the standard deviation of the mean.

## Data Availability

Original data is available from the corresponding author upon request.

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
