# Peer review of "Half-Life Extension and Biodistribution Modulation of Biotherapeutics via Red Blood Cell Hitch-Hiking with Novel Anti-Band 3 Single-Domain Antibodies"

_ijms, 2022, doi:10.3390/ijms24010475_

Round 1
Reviewer 1 Report
Revision of the review : “Half-life extension and biodistribution modulation of biotherapeutics via red blood cell hitch-hiking with novel anti-band 3 single-domain antibodies”.
The article describes the selection of sdAbs against anti-band 3 protein (b3p) in the RBC cell surface and it association to anti-TNF, generating a bispecific antibody. The evaluated the half-life of these antibodies and its biodistribution in mice, upon labelling with Iodine-125.
The article is interesting for the scientific community however major revisions need to be addressed before its publication.
Major revisions:
- The results of the selection of the sdAbs against b3p needs to be better represented. The authors should show the western blot of the purified sdAbs ( and how the selected sdAbs were labelled with Iodine-125. How stable is the labelling, is it affecting the binding to the b3p.
- The specificity of the selected sdAbs (e.g. against negative b3p cell) needs to be addressed.
- Can you please show some images of the biodistribution of the sdAbs in the mice and tissues (histology).
- Can you please also indicate how the dosage of the selected sdAbs labelled with Iodine-125 was determined to be administrated in the animal.
- Line 317- To further analyse the safety of the treatment other organs damage markers should be assessed, such as ALT, AST, AP, total leukocytes among others and histology of the organs.
Methods section:
- Please indicate how the sdAbs were labelled with I-125
- Line 75- Can you please indicate how it was verified that the B3p protein was correctly isolated?
- Please include in the methods how the Anti-RBC-alfaTNF Fusion Bispecific Antibody was generated.
Minor revisions:
Line 75 - Please indicate what buffer was used for the washes?
Line 102- Please Indicate what is PBST?
Line 143- Provide more details about how the SDS-PAGE was performed.
Line 146- Please indicate how the RBCs were isolated?
Line 157- Please explain better how the binding affinity to the RBCs was determined (include formulas).
Line 197- Please Indicate the wash buffer used.
Figure 1A- Can you please indicate what might be the other bands in the gel?
Line 226- Please provide more details about how the binding affinity was determined.
Figure 2- The figure 2B is cut, please correct it.
Line 285- Can you please provide the western blot of the purified constructs?
Author Response
The article describes the selection of sdAbs against anti-band 3 protein (b3p) in the RBC cell surface and it association to anti-TNF, generating a bispecific antibody. The evaluated the half-life of these antibodies and its biodistribution in mice, upon labelling with Iodine-125.
The article is interesting for the scientific community however major revisions need to be addressed before its publication.
R: We greatly appreciate your thoughtful review.
Major revisions:
- The results of the selection of the sdAbs against b3p needs to be better represented. The authors should show the western blot of the purified sdAbs ( and how the selected sdAbs were labelled with Iodine-125. How stable is the labelling, is it affecting the binding to the b3p.
Response: As requested, we have added the radiolabelling protocol in lines 177-187. Purified sdAb is now shown in figure 4 (SDS-PAGE). Although we did not assay labelled sdAb for binding to b3p, the observed pharmacokinetic results (e.g., Figure 5) provide clear demonstration of altered disposition (i.e., half-life extension).
- The specificity of the selected sdAbs (e.g. against negative b3p cell) needs to be addressed.
Response: As requested, we have now discussed We have addressed binding selectivity in lines 427-431.
- Can you please show some images of the biodistribution of the sdAbs in the mice and tissues (histology).
Response: Unfortunately, imaging of biodistribution was not performed. We recognize that this evaluation may be of interest and, while beyond the scope of the present work, may be pursued in future research.
- Can you please also indicate how the dosage of the selected sdAbs labelled with Iodine-125 was determined to be administrated in the animal.
Response: The dose of 1 mg/kg was selected empirically; this low dosage is expected to be below levels required for B3p saturation.
- Line 317- To further analyse the safety of the treatment other organs damage markers should be assessed, such as ALT, AST, AP, total leukocytes among others and histology of the organs.
Response: Thank you for this comment. Detailed safety investigations are beyond the scope of this investigation, but will be the subject of future research.
Methods section:
- Please indicate how the sdAbs were labelled with I-125
Response: The radiolabeling protocol was added in lines 177-187.
- Line 75- Can you please indicate how it was verified that the B3p protein was correctly isolated?
Response: Methods of b3p isolation from RBCs are well-established in the literature. Assessments of B3p isolation are now described in lines 87-90.
- Please include in the methods how the Anti-RBC-alfaTNF Fusion Bispecific Antibody was generated.
Response: We have added the methods of how the bispecific antibodies were generated in lines 131-133. We also included an SDS-PAGE image of the purified bispecific construct in Figure 4.
Minor revisions:
Line 75 - Please indicate what buffer was used for the washes?
We have added this information in line 75.
Line 102- Please Indicate what is PBST?
R: We have added this information in line 105.
Line 143- Provide more details about how the SDS-PAGE was performed.
R: We have added a more detail of the SDS-PAGE protocol in lines 148-154.
Line 146- Please indicate how the RBCs were isolated?
R: We have added the method for isolation of mouse RBCs in lines 157-161.
Line 157- Please explain better how the binding affinity to the RBCs was determined (include formulas).
R: We have added this information in lines 171-176.
Line 197- Please Indicate the wash buffer used.
R: The wash buffer used was 1% BSA in PBS. We have added this information in line 228.
Figure 1A- Can you please indicate what might be the other bands in the gel?
R: The other bands are possibly the proteolytic fragments or multimers of b3p. We have added this information in Figure 1A.
Line 226- Please provide more details about how the binding affinity was determined.
R: We have added in the method section in lines 171-176.
Figure 2- The figure 2B is cut, please correct it.
R: Revised as requested.
Line 285- Can you please provide the western blot of the purified constructs?
R: Western blots, which may be less demonstrative of construct purity relative to SDS-PAGE, were not performed.
Reviewer 2 Report
One of the problems with small biologicals such as single-domain antibodies is rapid clearance from the circulation and inadequate delivery to the therapeutic target site. Binding to red blood cells is a logical approach to extending the half-life in the circulation and improving delivery to the target site, as an alternative to PEGylation to increase molecular size. Binding to IgG or albumin has been explored previously. The authors selected a highly expressed protein on the RBC membrane and directed sd Abs against it. Using phage display, 9 clones were identified, of which 2 showed greatest promise in RBC binding studies with high affinity for human RBCs but some affinity for mouse cells. With affinity maturation, greater mouse RBC binding was achieved. These targeting moieties were then coupled to anti-TNF-alpha for pharmacokinetic studies, wherein the half-life of the fusion protein was ~50 times longer than native anti-TNF-alpha.
This is a well written paper with promising results. I have only a few minor comments.
MINOR
Line 161. Says “iodinated”, presumably iodine-125 as later in line 190, but the radioisotope should be specified.
Line 163. It would be helpful to state the number of blood samples and latest time point. It will be evident later in the results but should be here as well.
TYPOS ETC
Line 232. Should be superscript(125)I. Also line 247, 265, 277, 301, and possibly others
Although the Journal reference format is somewhat flexible, it is unusual to cite only one author before “et al.” Also, there is a mix of full journal titles and abbreviations
Author Response
Comments and Suggestions for Authors
One of the problems with small biologicals such as single-domain antibodies is rapid clearance from the circulation and inadequate delivery to the therapeutic target site. Binding to red blood cells is a logical approach to extending the half-life in the circulation and improving delivery to the target site, as an alternative to PEGylation to increase molecular size. Binding to IgG or albumin has been explored previously. The authors selected a highly expressed protein on the RBC membrane and directed sd Abs against it. Using phage display, 9 clones were identified, of which 2 showed greatest promise in RBC binding studies with high affinity for human RBCs but some affinity for mouse cells. With affinity maturation, greater mouse RBC binding was achieved. These targeting moieties were then coupled to anti-TNF-alpha for pharmacokinetic studies, wherein the half-life of the fusion protein was ~50 times longer than native anti-TNF-alpha.
This is a well written paper with promising results. I have only a few minor comments.
R: The reviewer's comments are greatly appreciated.
MINOR
Line 161. Says “iodinated”, presumably iodine-125 as later in line 190, but the radioisotope should be specified.
R: Revised as requested.
Line 163. It would be helpful to state the number of blood samples and latest time point. It will be evident later in the results but should be here as well.
R: Thank you for the input. We have added the time-points for blood collection in lines 194-196.
TYPOS ETC
Line 232. Should be superscript(125)I. Also line 247, 265, 277, 301, and possibly others
R: Revised as requested.
Although the Journal reference format is somewhat flexible, it is unusual to cite only one author before “et al.” Also, there is a mix of full journal titles and abbreviations
R: Revised as requested.
Round 2
Reviewer 1 Report
The authors reply to my comments, and made the needed correction. Minor correction are still required, such as the format of some of the figures. In the figure 1 please indicate the other values of the MW similarly to figure 4.
Author Response
Comments and Suggestions for Authors
The authors reply to my comments, and made the needed correction. Minor correction are still required, such as the format of some of the figures. In the figure 1 please indicate the other values of the MW similarly to figure 4.
Response: Thank you for the comment. As requested, we have added indication of the molecular weights of ladder proteins to Figure 1.